# Induction of labour for predicted macrosomia: study protocol for the 'Big Baby' randomised controlled trial

Lauren Jade Ewington ©,[1,2] Jason Gardosi,[3] Ranjit Lall,[4] Martin Underwood ©,[4,5] Joanne D Fisher,[4] Sara Wood,[4] Ryan Griffin,[4] Kirsten Harris,[4] Debra Bick ©,[4] Katie Booth,[4] Jaclyn Brown,[4] Emily Butler,[3] Kelly Fowler,[3] Mandy Williams,[3] Sanjeev Deshpande,[6] Adam Gornall,[6] Jackie Dewdney,[7] Karen Hillyer,[7] Simon Gates,[8] Ceri Jones,[5] Hema Mistry ©,[4] Stavros Petrou,[9] Anne-Marie Slowther,[10] Adrian Willis,[4] Siobhan Quenby[1,2,5]

For numbered affiliations see end of article.

**Correspondence to**
Professor Siobhan Quenby;
S.Quenby@warwick.ac.uk

## ABSTRACT

**Introduction** Large-for-gestational age (LGA) fetuses have an increased risk of shoulder dystocia. This can lead to adverse neonatal outcomes and death. Early induction of labour in women with a fetus suspected to be macrosomic may mitigate the risk of shoulder dystocia. The Big Baby Trial aims to find if induction of labour at $38^{+0}$–$38^{+4}$ weeks' gestation, in pregnancies with suspected LGA fetuses, reduces the incidence of shoulder dystocia.

**Methods and analysis** The Big Baby Trial is a multicentre, prospective, individually randomised controlled trial of induction of labour at $38^{+0}$ to $38^{+4}$ weeks' gestation vs standard care as per each hospital trust (median gestation of delivery $39^{+4}$) among women whose fetuses have an estimated fetal weight >90th customised centile according to ultrasound scan at $35^{+0}$ to $38^{+0}$ weeks' gestation. There is a parallel cohort study for women who decline randomisation because they opt for induction, expectant management or caesarean section. Up to 4000 women will be recruited and randomised to induction of labour or to standard care. The primary outcome is the incidence of shoulder dystocia; assessed by an independent expert group, blind to treatment allocation, from delivery records. Secondary outcomes include birth trauma, fractures, haemorrhage, caesarean section rate and length of inpatient stay. The main trial is ongoing, following an internal pilot study. A qualitative reporting, health economic evaluation and parallel process evaluation are included.

**Ethics and dissemination** The study received a favourable opinion from the South West— Cornwall and Plymouth Health Research Authority on 23/03/2018 (IRAS project ID 229163). Study results will be reported in the National Institute for Health Research journal library and published in an open access peer-reviewed journal. We will plan dissemination events for key stakeholders.

**Trial registration number** ISRCTN18229892.

## INTRODUCTION

Shoulder dystocia occurs when an infant's head has been delivered vaginally and the shoulder becomes stuck behind a woman's pubic bone. This can lead to maternal and fetal complications. Maternal complications include haemorrhage, third-degree and fourth-degree perineal tears and psychological sequelae. Infant complications include fractures of the clavicle and humerus, brachial plexus injury, hypoxic ischaemic encephalopathy and death.[1–3] Shoulder dystocia and its complications are common indications for litigation in obstetrics with settlements dealt with by the UK NHS Litigation Authority (now called NHS Resolution) from 250 cases between 2000 and 2010 costing over £100 million.[4]

### STRENGTHS AND LIMITATIONS OF THIS STUDY

⇒ This is the largest trial assessing if induction of labour decreases the incidence of shoulder dystocia in women with a suspected large-for-gestational (LGA) age fetus.

⇒ The main trial is currently open to recruitment, following a successful internal pilot study. The trial includes qualitative reporting, and health economic and process evaluations.

⇒ Women declining randomisation and opting for an elective caesarean section can consent to participate in a parallel cohort study to collect maternal and neonatal health outcomes.

⇒ Recruitment is challenging as women and clinicians often have a preference regarding timing and mode of birth and decline randomisation. Therefore, it is unclear if the women randomised into the trial are representative of the population.

⇒ Currently in the UK there is no guidance on the management of suspected LGA pregnancies, meaning the gestation of delivery of the standard care group is varied. Ongoing analysis of data from participants already involved shows the median gestation of delivery is $39^{+4}$ weeks' gestation.

Fetal macrosomia is a well-described risk factor for shoulder dystocia.[5] This is variably defined as a neonatal birth weight >4.0 kg or 4.5 kg, or >90th customised or non-customised fetal weight centile. Preventative measures start with antenatal awareness of risk factors including fetal growth and size, maternal obesity and diabetes.

Earlier delivery is likely to reduce the birth weight of the infant and mitigate the main risk factor for shoulder dystocia. However, it is uncertain whether this strategy would work to reduce shoulder dystocia and its associated complications, and what effect this might have on caesarean section rates and maternal complications after delivery. Research into prevention by induction is timely, in light of conflicting messages. The Royal College of Obstetricians and Gynaecologists (RCOG) does not currently recommend induction of labour for women with a suspected macrosomic fetus in the absence of diabetes.[6] However, two systematic reviews and meta-analyses found that induction of labour reduced the risk of shoulder dystocia in women who had a macrosomic fetus.[7 8] Both reviews were largely based on the 2015 randomised controlled trial by Boulvain and colleagues of 822 pregnancies with a fetus with an estimated weight greater than the 95th centile.[9] While inducing labour may reduce the risk of shoulder dystocia, it has not been shown to decrease adverse neonatal sequelae and induction is associated with a marginal increased risk of operative delivery.[10]

The management of large-for-gestational age (LGA) and macrosomic pregnancies in obstetrics was the focus of a landmark legal case heard by the UK Supreme Court in 2014.[11] Mrs Montgomery had type 1 diabetes and had a macrosomic baby, she was concerned about delivering her baby vaginally, but was not adequately informed of the risk of shoulder dystocia. During the delivery, shoulder dystocia occurred leading to a 12 min delay in delivering the infant's body. Her son suffered from hypoxic ischaemic encephalopathy. A case was made that as Mrs Montgomery was not adequately informed of the risk of shoulder dystocia and its associated complications, and the alternative modes of delivery, namely caesarean section, she could not make a well-informed decision about the delivery of her son, therefore there was negligence in consent. After failed appeals at the Court of Session and the Inner house the case was finally heard at the UK Supreme court. The Supreme Court judgement in this case highlighted the obligation of clinicians to explain the risks and benefits of all treatment options, including that of no treatment, to women for them give a valid consent. It is therefore imperative to have robust evidence from randomised controlled trials on which to base these discussions. An investigation into the value of induction to reduce the incidence of shoulder dystocia in women with a suspected macrosomic fetus will give women and clinicians the information they need in planning their mode of delivery.

The research question is 'does induction of labour at $38^{+0}$ to $38^{+4}$ weeks' gestation, in pregnancies with suspected LGA fetuses, reduce the incidence of shoulder dystocia?'.

This manuscript describes the trial design, setting, participants and recruitment, the intervention and control groups, randomisation, outcome measures, sample size, ethical considerations and dissemination. A separate manuscript will detail the statistical analysis plan, trial process evaluation and health economic analysis plan.

## STUDY OBJECTIVES
### Primary objective
The primary objective is to determine the effectiveness of induction of labour at $38^{+0}$ to $38^{+4}$ weeks' gestation in reducing the incidence of shoulder dystocia in suspected LGA fetuses.

### Secondary objective
Secondary objectives are to collect comparative data on intrapartum, perinatal, infant, maternal obstetric and long-term maternal outcomes. We will collect comparative data on maternal perceptions of their labour/birth care and physical and psychological health at 2 and 6 months postnatally. We will report composite outcomes for intrapartum birth injury, prematurity associated problems and maternal intrapartum complication.

## METHODS AND ANALYSIS
This protocol manuscript was written in concordance with the Standard Protocol Items: Recommendations for Interventional Trials guidelines.[12]

### Trial design
The Big Baby Trial is a multicentre, prospective, individually randomised controlled trial of induction of labour at $38^{+0}$ to $38^{+4}$ weeks' gestation versus standard care of fetuses that are LGA according to ultrasound scan at $35^{+0}$ to $38^{+0}$ weeks' gestation. Our definition of LGA is an estimated fetal weight >90th customised fetal weight centile using the woman's own customised Gestation Related Optimal Weight (GROW) chart.[13] These charts provide the standard for assessment of fetal growth and newborn size, are recommended by RCOG Green Top Guidelines[14] and are in use in approximately 76% of NHS Trusts and Health Boards. GROW charts adjust for maternal height, weight in early pregnancy, parity, ethnic origin and gender where known. Pathological variables such as diabetes and smoking are not adjusted for.[13 15] The GROW 90th customised centile identifies more babies at risk of adverse outcomes than LGA by conventional standards.[16–19] Furthermore, GROW has been shown to be a better predictor of shoulder dystocia than the UK-WHO birth weight standard.[20]

There is a parallel cohort study for women who decline randomisation but wish to participate in research. This cohort includes two subgroups. The first is women who

request a planned caesarean section. The second is women who request to be delivered by early induction of labour or expectant management. The primary objective of the cohort study is to provide comparative data on those who choose planned caesarean section and confirm generalisability of the baseline data and primary outcome with the main trial.

The trial is conducted and managed by the Warwick Clinical Trials Unit and sponsored by the University Hospitals Coventry and Warwickshire NHS Trust. Funding is provided by the National Institute for Health Research (NIHR) following a commissioned call from the Health Technology Assessment Programme (HTA study reference 16/77/02). The trial is being conducted in accordance with the principals of the Declaration of Helsinki and Good Clinical Practice (GCP).

## Trial setting

Although we initially planned to recruit from 60 NHS Trusts over the course of the trial to enable us to enhance recruitment, this approach has changed. We now aim to recruit 80 NHS Trusts across the UK that use customised GROW charts. Staff participating in the trial must demonstrate and document a willingness to comply with the protocol, the principles of GCP and regulatory requirements. Furthermore, they must be prepared to participate in training and adhere to the protocol.

## Participants and recruitment
### Inclusion criteria
The study participants are women aged ≥18 years with a fetus above the 90th customised GROW fetal weight centile on ultrasound scan at $35^{+0}$ to $38^{+0}$ weeks' gestation with a cephalic presentation.

### Exclusion criteria
Box 1 lists the exclusion criteria for the study.

---

### Box 1   Exclusion criteria

1. Multiple pregnancy.
2. Pregnancy with a breech or transverse lie position.
3. Contra-indication to induction of labour.
4. A fetus with a known serious abnormality.
5. A home birth or elective caesarean section already planned.
6. A caesarean section or induction indicated due to other health conditions such as cardiac disease or hypertensive disorders.
7. Women taking medications and/or insulin therapy for diabetes or gestational diabetes (women with these conditions who are not taking medication are eligible).
8. A current diagnosis of a major psychiatric disorder requiring antipsychotic medication.
9. A previous stillbirth or neonatal death ≤28 days.
10. A current intrauterine fetal death.
11. Prisoners.
12. Women unable to give informed consent for example, learning or communication difficulties that prevent the understanding of the information provided.

---

## Recruitment

Figure 1 describes the pathway women will take through the trial and the expected number of women at each stage. Women are identified based on an ultrasound scan, performed either as part of serial fetal growth assessment or for a different indication. If the fetus has an estimated fetal weight >90th customised centile from $28^{+0}$ to $38^{+0}$ weeks' gestation, the woman can be approached and offered information about the study. Women are informed of the risks and benefits of participating and the possible risks and benefits of other delivery options. These can be found in the participant information sheet (online supplemental material). The participant information sheet and participant consent form have been assessed for clarity by the Plain English Campaign and a Crystal Mark obtained for these. By approaching women from $28^{+0}$ weeks' gestation, they have time to consider their participation, ask questions to healthcare professionals and discuss the trial with their family and friends.

The obstetrician, or consultant midwife in charge of the woman's care is asked to provide 'obstetric confirmation', to confirm they agree for their patient to participate in the trial and receive either induction of labour or standard care. This confirmation must be completed before randomisation. To be eligible a confirmatory ultrasound scan must be performed between $35^{+0}$ and $38^{+0}$ weeks' gestation. If the fetus has an estimated fetal weight >90th customised GROW centile during this gestation interval and fulfils the other eligibility criteria, the woman can participate in the trial.

## Intervention and control
### Intervention
Data from the West Midlands Perinatal Episode Electronic Record database of 161 936 pregnancies found that the median length of pregnancy for LGA fetuses was $39^{+4}$ weeks' gestation (277 days). We further ascertained that the weekly increment of fetal weight gain in LGA pregnancies is approximately 200 g. In the trial conducted by Boulvain and colleagues, the difference in fetal weight between the induction and expectant management groups was 287 g.[9] Based on this, we expect that for a difference of 300 g between the intervention and control arms, an interval of 1.5 weeks is required. Therefore, the intervention window for induction of labour is set at $38^{+0}$ to $38^{+4}$ weeks' (266–270 days) gestation. This will ensure an approximate average of eleven days separation in gestation days between groups. Induction prior to this window may decrease the risk of shoulder dystocia but would increase the risk of neonatal complications.[21–23] The method of induction is by the usual practice at the participating site Trust.

### Control
The control is standard care. In the UK there is no guidance on mode and timing of birth in LGA pregnancies, with practice varying from hospital to hospital and clinician to clinician. Standard care for this trial is what is

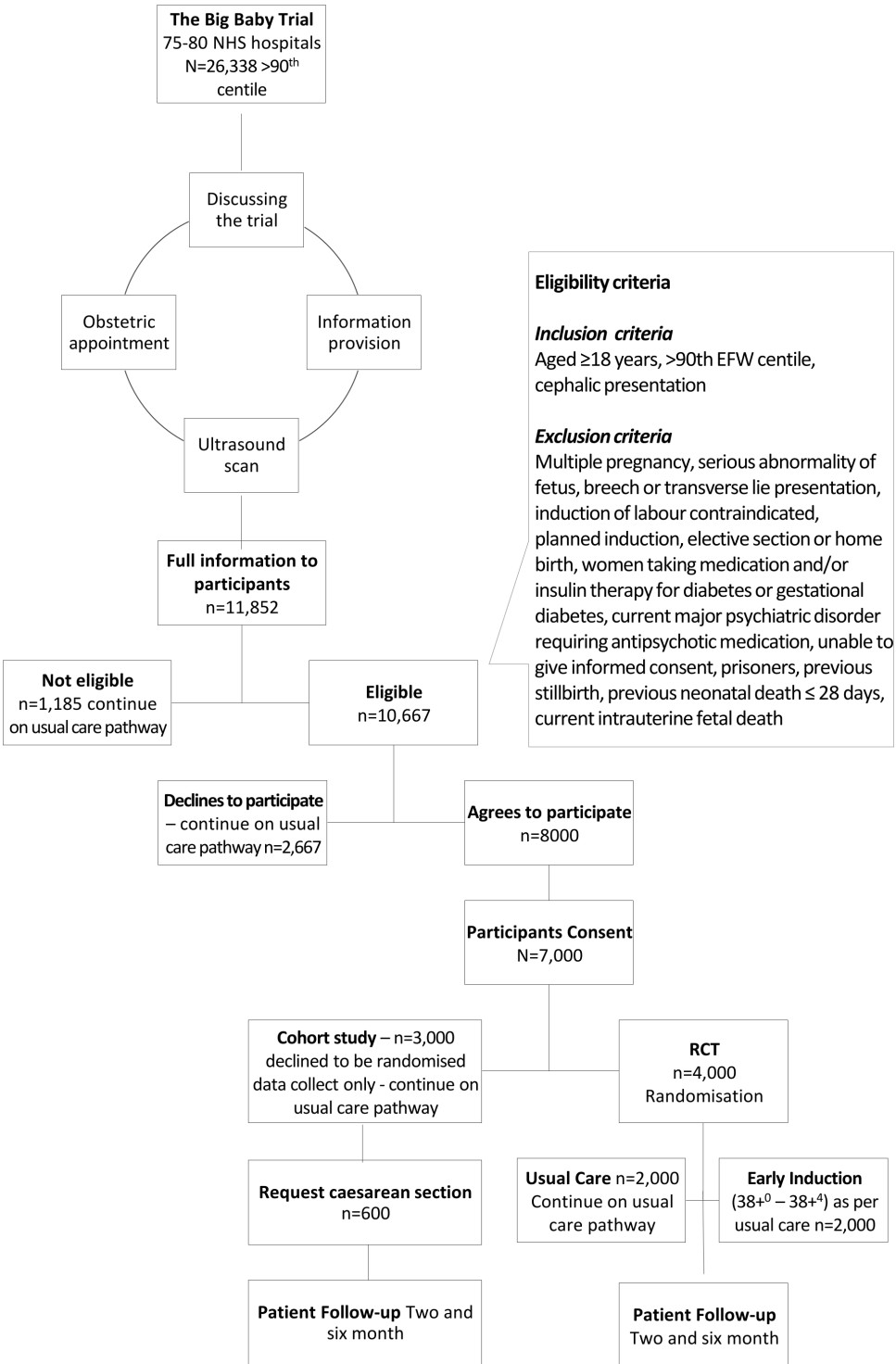

**Figure 1** Trial flow diagram with expected numbers of participants.

provided by that hospital. The trial data monitoring and ethics committee (DMEC) continue to review the gestation of delivery of the standard care arm and so far, the median gestation of birth in the standard care arm is $39^{+4}$ weeks' gestation.

## Outcome measures
### Primary outcome
The primary outcome measure is the incidence of shoulder dystocia, defined by the RCOG as 'a vaginal cephalic delivery that requires additional obstetric manoeuvres to deliver the fetus after the head has delivered and gentle traction has failed'.[6] These data are being extracted from clinical notes.

**Table 1** Secondary outcomes

| Maternal peri partum | Fetal peri partum | Neonatal |
|---|---|---|
| Duration of hospital stay prior to delivery | Time recorded between delivery of the head and delivery of the body | Neonatal death |
| Duration of hospital stay after delivery | Time in labour ward | Birth weight |
| Mode of delivery | Time from commencement of the active second stage of labour until fetal expulsion | Gestation at birth |
| Perineal tears | Stillbirth | Apgar score at 5 min |
| Vaginal and cervical lacerations | | Fractures |
| Primary postpartum haemorrhage | | Brachial plexus injury |
| Clinician defined sepsis | | Clinician defined sepsis |
| Fever >38.0°C given antibiotics | | Given antibiotics |
| Retained placenta | | Admission to the neonatal unit (intensive, special or transitional care) |
| Uptake of breast feeding | | Duration of hospital stay |
| Hospital readmission within 30 days of postnatal inpatient discharge | | Hypoxic ischaemic encephalopathy |
| Death | | Use of phototherapy |
| | | Respiratory morbidity |
| | | Hypoglycaemia |

As the sites are unblinded, all delivery notes are reviewed by an independent expert panel to confirm if shoulder dystocia has occurred. The independent panel consists of a senior obstetrician, a senior neonatologist, a senior midwife and a trainee obstetrician. Delivery notes are anonymised. The independent panel is blind to the trial allocation. Two panel members review each set of notes and categorise the notes into: (1) delivered by caesarean section; (2) no shoulder dystocia; (3) shoulder dystocia; or (4) needs more clarification. Where more clarification is needed, additional information is being sought from trial sites. If there is discrepancy between panel members, the entire panel discusses the case until a consensus decision is made.

### Secondary outcomes

The secondary outcomes are grouped into maternal peripartum, fetal peripartum, neonatal outcomes and longer-term outcomes. The secondary outcomes captured from the admission for delivery are defined in table 1.

Randomised participants and participants in the cohort study opting for an elective caesarean section are asked to complete questionnaires at 2 and 6 months post partum. The outcomes for the infants are assessed according to the proportion under specialist medical care at 2 months for a problem related to intrapartum experience, maternal report of infant health concerns at 6 months, in hospital healthcare costs and hospital readmission within 30 days of postnatal inpatient discharge. Responses from these questionnaires identify infants who have sustained a potential birth-related injury. Relevant data related to the injury are being requested from sites and an independent

adjudication committee will classify these as delivery/not delivery related. This will be undertaken by the same independent adjudication committee that is to review the delivery notes. Box 2 details the longer-term maternal and neonatal outcomes.

The three composite outcomes are:
► Peripartum birth injury: includes one or both of fractures or brachial plexus injury.
► Prematurity associated problems which include one or more of phototherapy, clinician defined sepsis before discharge from hospital or respiratory support.
► Maternal peripartum complications which include one or more of third and fourth degree perineal tears, vaginal/cervical lacerations, clinician defined

---

**Box 2   Longer-term maternal and neonatal outcomes**

**Longer-term outcomes**
⇒ Maternal experience (six simple questions) at 2 months.[27]
⇒ Duration of exclusive breast feeding at 2 and 6 months.
⇒ Health-related quality of life (EQ-5D-5L) at 2 and 6 months.[24]
⇒ Edinburgh Postnatal Depression Scale score at 2 and 6 months.[25]
⇒ Impact of Events Scale at 2 months.[28]
⇒ Postpartum bonding questionnaire at 2 months.[29]
⇒ Maternal report of infant health at 2 and 6 months.
⇒ Urinary incontinence ICIQ-UI short form at 2 and 6 months.[26]
⇒ Faecal incontinence at 2 and 6 months.
⇒ Sexual function at 6 months.
⇒ Maternal and infant death at 6 months from HES-ONS linked mortality data.
⇒ Participants health resource used for the economic analysis for mother and baby at 2 and 6 months.

sepsis before discharge from hospital or primary post-partum haemorrhage.

## Sample size

The true incidence of shoulder dystocia in women with a fetus >90th customised GROW centile is unknown. In the trial by Boulvain and colleagues on suspected macrosomia, the incidence of shoulder dystocia, defined as 'difficulty with delivery of the shoulders not resolved by McRoberts manoeuvre', in the control arm was 16/411 (3.9%).[9] In the Big Baby Trial, we have used a similar definition of shoulder dystocia, and have estimated the incidence of shoulder dystocia in the control group to be 4%. Boulvain *et al* found a relative risk for significant shoulder dystocia in the intervention group to be 0.32 (95% CI 0.12 to 0.85).[9] Considering this, we have set the effect size to 50% reduction in the primary outcome to 2%. This reduction is considered clinically worthwhile. To achieve a 50% reduction in the primary outcome at a 5% significance level with 90% power, 1626 women would need to be allocated to each arm, with a sample size of 3252 women.

The sample size for this trial has been increased from 3252 by 23% to 4000. This is to allow for some women giving birth prior to the intervention, and to account for uncertainty in the event rate in the control group. In the trial by Boulvain and colleagues, 31/408 women (7.6%) gave birth prior to the intervention.[9] The increase in the sample size also takes into account the unknown incidence of the primary outcome, an expected small loss of primary outcome, and any effect of clustering at site—although an unpublished analysis of national Growth Assessment Protocol data by the Perinatal Institute indicated the intra-cluster correlation coefficient for being LGA to be <0.00055, suggesting that any effect will be negligible.

The trial DMEC is presented with a closed and open report of the data every 6 months of the study. A key event analysis was undertaken once primary outcome data were collected for 1000 participants, given the uncertainty in the sample size estimate. The DMEC was asked to advise if a sample size adjustment was required based on the incidence of shoulder dystocia in the control arm. These data were available on 5 February 2020 and were considered by the DMEC who were unanimous in their satisfaction of the original planned target and recommended that the trial continues to recruit the planned 4000 women.

## Internal pilot, process evaluation and qualitative interviews

Recruitment was assessed when ten sites had been recruiting for 3 months. A formative process evaluation was undertaken to assess barriers to recruitment of sites and participants and barriers to follow-up. This included interviews with ten clinicians to explore adherence to study protocol, impact on workload and impact of the trial on the woman's decision-making process. Feedback from the pilot study and process evaluation allowed us to run seamlessly into the main study. This will be described in a further manuscript.

## Randomisation

Randomisation is provided by Warwick Clinical Trials Unit using an online web application or telephone. Women are randomised using minimisation, balancing site, fetal weight centile (≤95th or >95th estimated fetal weight centile) and maternal age (≤35 or >35 years of age). To ensure allocation concealment, randomisation only takes place once all the baseline data have been collected. Women are randomised to either booking of induction of labour between 38$^{+0}$ and 38$^{+4}$ weeks' gestation or to standard care. Women are immediately informed of the allocation.

## Data collection

Anonymised data are entered into a secured password protected trial database, developed by the programming team at Warwick Clinical Trials Unit, either at site or by the Warwick Clinical Trials Unit. Participants are identified by a unique study number. All data are stored securely and held in accordance with the relevant UK data protection legislation.

The baseline data collected are maternal height, weight, age, parity, ethnic origin, obstetric history, current obstetric history, tobacco use and use of antenatal corticosteroids. Women are asked to complete the EQ-5D-5L health-related quality of life questionnaire,[24] Edinburgh Postnatal Depression Scale score,[25] urinary incontinence ICIQ-UI short form[26] and questions on faecal incontinence and sexual function at baseline.

The fetal and neonatal outcomes collected are detailed in table 1. In addition, we are collecting data on the proportion of infants under specialist medical care at 2 months for a problem related to intrapartum experience, a maternal report of infant health at 6 months and in-hospital costs. The maternal outcomes collected are described in table 1. Longer-term maternal outcomes to be collected are described in box 2.

Follow-up questionnaires are sent to participants at 2 and 6 months post partum. We check the hospital electronic record for notification of a neonatal death in all infants participating in the study who were discharged home, prior to sending the follow-up questionnaires. All study related data are stored in accordance with all applicable regulatory requirements and access is restricted to authorised personnel. Trial records and associated documentation will be archived for 25 years for the randomised participants and 10 years for the cohort participants.

For the parallel cohort we collect the same baseline data as the randomised controlled trial. For women requesting a planned caesarean section we collect the same maternal, neonatal and infant outcomes as the randomised controlled trial. There is a limited data collection for women in the cohort study who request induction or standard care. Women have been consented to be approached for longer-term follow-up.

## Data analysis

All analyses will be by intention to treat at the time of randomisation. The primary analysis will compare the incidence of shoulder dystocia between the intervention and control groups. The comparison will be made using logistic regression models both unadjusted and adjusted for appropriate covariates. Other secondary binary outcomes will be assessed in a similar way. Continuous outcomes will be analysed using linear regression models; both adjusted and unadjusted analyses will be computed. A description of the data analyses are described in a further manuscript.

## ETHICS AND DISSEMINATION
### Ethical conduct of the trial

The trial complies with all UK legislation and Warwick Clinical Trials Unit standard operating procedures. Health Research Authority approval and NHS Trust R&D approval was obtained before participants were enrolled in the trial.

A key ethical challenge in this trial was to ensure that robust informed consent was obtained from participants. The trial requires women to consent to being randomised to a specific management pathway for the birth of their child rather than the standard clinical practice of a shared decision-making process with their clinician. It was therefore an imperative to provide the best possible information to women about the risks and benefits of all management options so they could make an informed decision about trial participation in the wider context of decision-making about their clinical care. In developing our information materials and consent processes we were guided by the standard set by the Supreme Court judgement in Montgomery.[11] The key steps we took to develop the information and consent processes were:

► A review of all relevant literature from the RCOG, National Institute for Health and Care Excellence and other published works.
► Development of participants facing materials with the patient and public involvement representatives.
► A thorough peer-review of all participant facing materials by obstetricians.
► The inclusion of a cohort group to respect the woman's preferred choice.

### Adverse event management

Adverse events are being collected from the time of randomisation until delivery. Serious adverse events (SAEs) are being collected from the time of randomisation until 30 days after initial discharge following delivery. Adverse events and SAEs are being identified when collecting outcome data or when completing the 2-month follow-up questionnaires.

For the trial only, adverse events affecting the woman or her baby which could be potentially related to the pregnancy, delivery or care of the neonate are being collected. Adverse events are being collected for all participants in

**Table 2** Serious adverse events that require immediate reporting for the woman and neonate

| Maternal serious adverse events | Neonatal serious adverse events |
|---|---|
| Maternal death | Stillbirth |
| Inpatient admission to intensive care and/or high dependency unit at any time during pregnancy/postnatal period | Infant death |
| Readmission to hospital within 30 days of initial postnatal discharge | Inpatient admission to the neonatal unit |
| Antenatal hospital admission not related to pregnancy | Inpatient readmission to hospital within 30 days of initial postnatal discharge* |
| Transfer out of the maternity unit for further inpatient care | |
| Inpatient admission to a mental health unit | |
| Symphysiotomy | |

*Except for respiratory tract infection, jaundice, urinary tract infection, weight loss lasting less than 5 days, reflux and constipation.

the randomised controlled trial and participants in the cohort study requesting an elective caesarean section.

SAEs are only being collected for participants in the randomised controlled trial and need to be reported to Warwick Clinical Trials Unit within 24 hours of the site being made aware of the event. Certain events that would meet the definition of SAEs are common in pregnancy and for this trial do not need to be reported as SAEs. These events are being reported in the trial case report forms and comparative rates will be monitored by the DMEC. SAEs that require immediate reporting for the woman and neonate are described in table 2.

For all SAEs a clinical assessment of causality is being made by a medical doctor as to whether the event is related to the booking of induction of labour. If the site or sponsor determine that there is a possible, probable or definite relationship to the intervention then an assessment of expectedness is completed. Related and unexpected SAEs are expedited to the Health Research Authority Research Ethics Committee, the sponsor and the chairs of the Trial Steering Committee and DMEC.

### Monitoring

All clinicians involved in obtaining consent are required to have completed GCP training. A programme of training is being delivered to all staff participating in the trial at site level. Data entered into the trial database are being checked for accuracy and completeness by Warwick Clinical Trials Unit in accordance with the trial data management plan. A risk assessment is being undertaken and forms the basis of the trial monitoring

plan. Following site initiation, the trial team is in regular contact with sites.

## Patient and public involvement

Karen Hillyer (Chair) and Jackie Dewdney (Board Member) of the Erb's Palsy group are actively involved in the planning and development of this trial. The Erb's Palsy group is a UK-based not for profit organisation which offers advice, support and information to families affected by Erb's Palsy. Karen and Jackie led on the development of all patient-facing materials. As coapplicants they are involved in all aspects of the trial and will help inform the interpretation of the final results and dissemination of findings.

## Progress so far

The trial started recruiting on 8 June 2018. As of 17 September 2021, there are 2261 randomised participants and 1566 cohort participants. Recruitment was paused on 23 March 2020 because of the COVID-19 pandemic. This restarted on a site-by-site basis depending on site capacity from 22 May 2020.

## Dissemination

The trial results will be reported in the NIHR journals library and published in an open access peer reviewed journal. Findings will be made available on the University of Warwick and Perinatal Institute websites. Abstracts will be submitted to major national and international conferences. Three dissemination events will be held for key stakeholders at the end of the trial. The trial will be reported in accordance with CONSORT guidelines. All publications will be submitted to the NIHR-HTA Programme for approval prior to submission for publication.

## Changes made since funding agreed

Since submission of the detailed project description to the NIHR-HTA some changes have been made to the protocol and agreed by the Trial Steering Committee, and DMEC. This section details the changes made and reasons for these.

Initially we predicted we would need 60 sites to reach our recruitment target. Over the course of the trial, it was evident this would need to be increased to 80 sites to enable us to improve recruitment and reach our target of 4000 women randomised in a timely manner. In the application to the NIHR-HTA we wanted to collect outcomes on women in the cohort study who had requested an elective caesarean section. It was decided by the Trial Management Group and Trial Steering Committee that this should be extended to include outcomes on women who decline randomisation but chose either to have an early induction of labour or expectant management. The objective of this group was to provide comparative data on those who choose the timing of the birth and to confirm generalisability of the baseline data and primary outcome. Women with a current intrauterine fetal death were added to the current exclusion criteria as it is

inappropriate to randomise these women and different plans would be made regarding their delivery. Prisoners were also added as a new exclusion criterion as there is a different ethical framework for their participation in medical research.

In the initial application to the NIHR-HTA we suggested that SAEs will be reported for any incidences of stillbirth, maternal death, serious intrapartum injury to the fetus or any other event that could be classified with similar severity. Once the trial had started recruiting a substantial number of SAEs were being reported that were classified as outcomes for the trial. Therefore, more formal guidance was formulated to avoid repetition in the data collection for events that did not meet the definition of SAE and to give clear instructions to the sites about what needed to be reported.

As a consequence of ongoing COVID-19 risk we are implementing a new consent process to allow for remote electronic consent rather than all consent being taken in person.

**Author affiliations**
[1]Biomedical Sciences, University of Warwick Faculty of Medicine, Coventry, UK
[2]Women's and Children's, University Hospitals Coventry and Warwickshire NHS Trust, Coventry, UK
[3]Perinatal Institute, Birmingham, UK
[4]Warwick Clinical Trials Unit, Warwick Medcial School, University of Warwick, Coventry, UK
[5]University Hospitals Coventry and Warwickshire NHS Trust, Coventry, UK
[6]Shrewsbury and Telford Hospital NHS Trust, Shrewsbury, UK
[7]Erb's Palsy Group, Coventry, UK
[8]Cancer Research UK Clinical Trials Unit, University of Birmingham, Birmingham, UK
[9]Nuffield Department of Primary Care Health Sciences, Oxford, UK
[10]Social Sciences and Systems in Health, University of Warwick, Coventry, UK

**Contributors** All authors read and approved the manuscript. All authors have contributed to the study design. SQ and JG are the cochief investigators and oversee the running of the study. MU input into all aspects of the study design and support in running the study. LJE is a Clinical Research Fellow and assisted with all aspects of the delivery of the interventions at site level. SW, KH, RG and JB managed the trial and data management. DB, EB, KF, SD, AG provided the clinician and midwifery input into the study. JDF carried out the process evaluation. KB, RL and SG were the statisticians for the study. JD, KH were the patient and public involvement representatives. SP and HM provided oversight of the health economic aspects of the study. A-MS was the ethicist for the study. AW and MW over sited the programming and database management and CJ was the sponsorship representative.

**Funding** This project is funded by the National Institute for Health Research, Health Technology Assessment (NIHR HTA) project number 16/77/02. The views and opinions expressed therein are those of the authors and do not necessarily reflect those of the HTA, NIHR, NHS or the Department of Health.

**Competing interests** JG is the director of the Perinatal Institute, a not for profit organisation, limited by guarantee, and a qualified provider of maternity support services to the NHS. It derives its income from some of its products and services, including the award-winning GAP programme mentioned in this protocol, through which they have been able to implement training, e-learning and protocols in the majority of Trusts and Health Boards in the UK. GAP includes the standardised, RCOG endorsed customised GROW charts which will be used to identify large-for-gestational age as the entry point for this trial. MU is chief investigator or coinvestigator on multiple previous and current research grants from the UK National Institute for Health Research, Arthritis Research UK and is a coinvestigator on grants funded by the Australian NHMRC. He is an NIHR Senior Investigator. He has received travel expenses for speaking at conferences from the professional organisations hosting the conferences. He is a director and shareholder of Clinvivo

Ltd http://www.clinvivo.com that provides electronic data collection for health services research. He is part of an academic partnership with Serco Ltd, funded by the European Social Fund, related to return to work initiatives. He is a coinvestigator on two NIHR funded studies receiving additional support from Stryker Ltd. He has accepted honoraria for teaching/lecturing from consortium for advanced research training in Africa. He was until March 2020 an editor of the NIHR journal series, and a member of the NIHR Journal Editors Group, for which he received a fee.

**Patient and public involvement** Patients and/or the public were involved in the design, or conduct, or reporting, or dissemination plans of this research. Refer to the Methods section for further details.

**Patient consent for publication** Not applicable.

**Provenance and peer review** Not commissioned; externally peer reviewed.

**ORCID iDs**
Lauren Jade Ewington http://orcid.org/0000-0003-0805-6845
Martin Underwood http://orcid.org/0000-0002-0309-1708
Debra Bick http://orcid.org/0000-0002-8557-7276
Hema Mistry http://orcid.org/0000-0002-5023-1160

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
