## [Reviewer comments · BMJ Open]

ARTICLE DETAILS

TITLE (PROVISIONAL)	Induction of Labour for Predicted Macrosomia: Study Protocol for the 'Big Baby' Randomised Controlled Trial
AUTHORS	Ewington, Lauren; Gardosi, Jason; Lall, Ranjit; Underwood, Martin; Fisher, Joanne; Wood, Sara; Griffin, Ryan; Harris, Kirsten; Bick, Debra; Booth, Katie; Brown, Jaclyn; Butler, Emily; Fowler, Kelly; Williams, Mandy; Deshpande, Sanjeev; Gornall, Adam; Dewdney, Jackie; Hillyer, Karen; Gates, Simon; Jones, Ceri; Mistry, Hema; Petrou, Stavros; Slowther, Anne-Marie; Willis, Adrian; Quenby, Siobhan

VERSION 1 – REVIEW

REVIEWER	Middleton, Philippa University of Adelaide
REVIEW RETURNED	23-Jan-2022

GENERAL COMMENTS	The present manuscript does not appear to reference the existing 2018 protocol and there do appear to be some differences between this and the current manuscript. I also think that it should be stated that recruitment has been completed - for example I was confused by the statement "the main trial ran seamlessly following an internal pilot study" until I accessed the ISRCTN records. Ref 10 has been misquoted - the Cochrane review does not report that labour is longer or more painful, or that there is an increase in operative births. Some more description of the choice of customised charts for LGA would be welcome.
---

REVIEWER	Elden, Helen Goteborgs Universitet, Health and Care Sciences
REVIEW RETURNED	10-Feb-2022

GENERAL COMMENTS	Thanks for the ability to review this study protocol for an interesting study, which will have impact for the management of labor for expected big babies. The parallel cohort study is a good initiative to get comparative data and confirm generalisability of the baseline data and primary outcome with the main trial. However, there are some information that needs to be clarified. Line 5 to 16: Limitations of the study are missing, please add. Line 55: There are several studies showing that induction of labour does not lead to increased risk for operative delivery ie: Walker et al. N Engl J Med 2016; 374:813-822 DOI: 10.1056/NEJMoa1509117 (35-39 study), INDEX study Keulen et al. BMJ 2019 and SWEPIS, Wennerholm et al. BMJ 2019. Also, Wennerholm et al. showed a significant higher rates of LGA babies
--

	but no increased risks for shoulder dystocia. Line 13: What is standard care for management of labor with expectant large babies? IOL at 39+4 days? Line 16: Please add the information given about risks of participating in the trial that are given to eligible women. Are all midwives, obstetricians and registrar doctors at the participating centers trained in McRobert maneuver? It is important to monitor mode of delivery. However, do you also monitor the management of the second stage of labor? Eg. birth position, use of oxytocin, technique to deliver the baby during one or several contractions, which can have impact on the risks of shoulder dystocia, please clarify. The description of the statistics is essential to judge if they are appropriate, and so is a CONSORT Flow-chart for an RCT, please add.
--	--

VERSION 1 – AUTHOR RESPONSE

Response to Reviewer 1’s Comments

The present manuscript does not appear to reference the existing 2018 protocol and there do appear to be some differences between this and the current manuscript.

Thank you for highlighting the discrepancies between the ISRCTN registry and the current manuscript. We have requested an update to the trial registry. We have added in a section on page 19 detailing the changes we have made to the trial protocol since the funding was agreed.

I also think that it should be stated that recruitment has been completed - for example I was confused by the statement "the main trial ran seamlessly following an internal pilot study" until I accessed the ISRCTN records.

Thank you for noting that this was not clear. Due to delays with COVID-19 the trial is currently recruiting. We have updated the ISRCTN records and made this clearer in the strengths and limitations section.

Ref 10 has been misquoted - the Cochrane review does not report that labour is longer or more painful, or that there is an increase in operative births.

Our apologies for misquoting the Cochrane review. We have changed this in the introduction sections to say induction of labour is associated with a marginal increased risk of operative delivery.

Some more description of the choice of customised charts for LGA would be welcome.

We have included a description of the customised charts and the variables that they do and do not adjust for.

Response to Reviewer 2’s Comments

Thanks for the ability to review this study protocol for an interesting study, which will have impact for the management of labor for expected big babies. The parallel cohort study is a good initiative to get comparative data and confirm generalisability of the baseline data and primary outcome with the main trial. However, there are some information that needs to be clarified.

Line 5 to 16: Limitations of the study are missing, please add.

Thank you for taking the time to review our manuscript and protocol paper. We have now added in two limitations, with are the difficulties with standard care and the difficulties with recruitment.

Line 55: There are several studies showing that induction of labour does not lead to increased risk for operative delivery ie: Walker et al. N Engl J Med 2016; 374:813-822

DOI: 10.1056/NEJMoa1509117 (35-39 study), INDEX study Keulen et al. BMJ 2019 and SWEPIs, Wennerholm et al. BMJ 2019. Also, Wennerholm et al. showed a significant higher rates of LGA babies but no increased risks for shoulder dystocia.

Thank you for highlighting these studies to us we have included a reference from the Cochrane review which showed a marginal increase in operative delivery and have changed the introduction to reflect this.

Line 13: What is standard care for management of labor with expectant large babies? IOL at 39+4 days?

Thank you for pointing this out. We have added in the abstract that this is according to hospital trust and have described this in further detail in the main manuscript about the variation in standard care across the UK and how our data monitoring committee are reviewing this regularly.

Line 16: Please add the information given about risks of participating in the trial that are given to eligible women.

We have included the patient information sheet as supplementary material which covers the risks and benefits of participating in the trial.

Are all midwives, obstetricians and registrar doctors at the participating centers trained in McRobert maneuver?

We have not included this in the manuscript as every clinical member of maternity staff at each hospital trust must undergo annual training on the PROMPT course. This includes the management of shoulder dystocia. This training is not given by the Big Baby team.

It is important to monitor mode of delivery. However, do you also monitor the management of the second stage of labor? Eg. birth position, use of oxytocin, technique to deliver the baby during one or several contractions, which can have impact on the risks of shoulder dystocia, please clarify. The description of the statistics is essential to judge if they are appropriate, and so is a CONSORT Flow-chart for an RCT, please add.

Thank you for your comments, the case report form for the trial collects data on the use of oxytocin, however does not collect data on birth position and technique to deliver the baby.

We have included a description of the statistics for the trial on line 363. A further manuscript will detail the full statistic analysis plan.

We have included the flow diagram.